# Differences in Toxic Response Induced by Three Variants of the Diarrheic Shellfish Poisoning Phycotoxins in Human Intestinal Epithelial Caco-2 Cells

**DOI:** 10.3390/toxins12120783

**Published:** 2020-12-08

**Authors:** Antoine Huguet, Olivia Drapeau, Fanny Rousselet, Hélène Quenault, Valérie Fessard

**Affiliations:** 1Toxicology of Contaminants Unit, Fougères Laboratory, French Agency for Food, Environmental and Occupational Health & Safety (ANSES), 35306 Fougères CEDEX, France; olivia.drapeau@hotmail.fr (O.D.); rousselet.fanny@laposte.net (F.R.); valerie.fessard@anses.fr (V.F.); 2Viral Genetic and Biosecurity Unit, Ploufragan-Plouzané Laboratory, French Agency for Food, Environmental and Occupational Health & Safety (ANSES), 22440 Ploufragan, France; helene.quenault@anses.fr

**Keywords:** phycotoxins, Caco-2 cells, transcriptomics, toxicity, okadaic acid, diarrheic shellfish poisoning

## Abstract

Diarrheic shellfish poisoning (DSP) is caused by the consumption of shellfish contaminated with a group of phycotoxins that includes okadaic acid (OA), dinophysistoxin-1 (DTX-1), and dinophysistoxin-2 (DTX-2). These toxins are inhibitors of serine/threonine protein phosphatases 1 (PP1) and 2A (PP2A), but show distinct levels of toxicity. Aside from a difference in protein phosphatases (PP) inhibition potency that would explain these differences in toxicity, others mechanisms of action are thought to be involved. Therefore, we investigated and compared which mechanisms are involved in the toxicity of these three analogues. As the intestine is one of the target organs, we studied the transcriptomic profiles of human intestinal epithelial Caco-2 cells exposed to OA, DTX-1, and DTX-2. The pathways specifically affected by each toxin treatment were further confirmed through the expression of key genes and markers of toxicity. Our results did not identify any distinct biological mechanism for OA and DTX-2. However, only DTX-1 induced up-regulation of the MAPK transduction signalling pathway, and down-regulation of gene products involved in the regulation of DNA repair. As a consequence, based on transcriptomic results, we demonstrated that the higher toxicity of DTX-1 compared to OA and DTX-2 was consistent with certain specific pathways involved in intestinal cell response.

## 1. Introduction

Phycotoxins, mainly produced by dinoflagellates, are natural metabolites that can accumulate in shellfish, especially in bivalves due to their high level of filter-feeding activity [1]. Among the lipophilic phycotoxins, okadaic acid (OA) and its main analogues, dinophysistoxin-1 (DTX-1) and dinophysistoxin-2 (DTX-2), are produced by certain species belonging to the genera *Prorocentrum* and *Dinophysis* [2]. Consumption of shellfish contaminated by OA and DTXs can lead to diarrheic shellfish poisoning (DSP), with clinical symptoms including nausea, vomiting, diarrhoea, and abdominal pain [3,4]. Although no human fatalities have been reported so far, the recurrent presence of these toxins in shellfish remains a public health concern [5].

OA and DTXs are inhibitors of serine/threonine protein phosphatases 1 (PP1) and 2A (PP2A) [6]. As PP1 and PP2A are involved in many cellular functions, their inhibition can affect several processes including cell cycle regulation, intracellular metabolism, cytoskeleton rearrangement, and gene expression [7,8,9]. Although OA, DTX-1, and DTX-2 have strong structural similarity, the number and position of the methyl groups have been shown to modulate the affinity for protein phosphatases (PP) [10,11], and therefore the inhibition potency (Figure 1). Concerning this inhibition potency for PPs, the IC_50_ for DTX-2 is the highest, while for OA and DTX-1 the response depends on the PP considered (IC_50_ OA > IC_50_ DTX-1 regarding PP2A, IC_50_ OA < IC_50_ DTX-1 regarding PP1) [6,12,13]. As a consequence, it has been reported that the toxicity of the three analogues differ (toxicity of DTX-1 > OA > DTX-2) in several studies performed in vitro with a range of cell lines [14,15,16], as well as in vivo in rodents [17,18,19]. Nevertheless, PP1 and PP2A inhibition is probably not the only process explaining the differences in toxicity, and the involvement of other mechanisms of action cannot be ruled out [3,20].

As studies exploring the comparison of toxicological mechanisms induced by OA, DTX-1, and DTX-2 are lacking, we investigated and compared the cellular and molecular mechanisms involved in the toxicity of these three analogues in undifferentiated human intestinal epithelial Caco-2 cells. The intestine is one of the target organs for these toxins and several publications have logically demonstrated the toxicity of OA, DTX-1, and DTX-2 on various intestinal cell lines. As the three analogues can affect cell cycle regulation, experiments were performed with undifferentiated (proliferating) Caco-2 cells rather than on differentiated (quiescent) cells. Moreover, undifferentiated Caco-2 cells showed higher sensitivity to toxins [15,21,22] compared to differentiated Caco-2 cells [23]. Therefore, we compared the transcriptomic profiles of Caco-2 cells exposed for 24 h to a single concentration of OA, DTX-1, and DTX-2 using a non-targeted approach with microarrays. The specific pathways identified for each toxin were further confirmed by evaluating the expression of selected genes, and by investigating toxicity markers with a multiplex approach (high content analysis, HCA).

## 2. Results

### 2.1. Cytotoxicity

Regardless of the toxin (OA, DTX-1, or DTX-2) tested, a concentration-dependent decrease in viability was observed in Caco-2 cells after 24 h of exposure (Figure 2A). No decrease in cell viability was observed up to 7 nM DTX-1 and up to 28 nM OA and DTX-2. Cytotoxicity reached 100% with 111 nM DTX-1, 442 nM OA, and 445 nM DTX-2. IC_50_ values were determined: 25 ± 4 nM for DTX-1, 75 ± 10 nM for OA, and 78 ± 12 nM for DTX-2. The IC_50_ of DTX-1 differed significantly from that of OA and DTX-2 (Figure 2B).

### 2.2. Transcriptomic Profile

We determined up- and down-regulated genes following treatment and selected those specific to each toxin. Following OA treatment, there were 36 and 18 specifically up and down-regulated genes, respectively, while they reached 745 and 572 specific to DTX-1, and 127 and 149 specific to DTX-2, respectively. The biological and molecular processes affected specifically by OA, DTX-1, and DTX-2 were addressed using the GoMiner application and the Database for Annotation, Visualization and Integrated Discovery (DAVID) Functional Annotation Tool.

For the genes specifically regulated by OA, no significant Gene Ontology (GO) term was depicted for the up-regulated genes, while 16 biological processes were significantly enriched for the down-regulated genes (Appendix A). These biological processes were mainly related to the regulation of cell proliferation and differentiation, and to vitamin metabolism. No enriched cellular component and no enriched molecular function could be detected. The additional analysis using the DAVID Functional Annotation Tool revealed no significant term for the down-regulated genes, and, for the up-regulated genes, only one significant term called the “Jak-STAT signalling pathway”, which is involved in different processes such as cell division and immunity (Appendix A). Three genes related to this signalling pathway belong to the OA up-regulated cluster.

On analysis of the 745 genes specifically up-regulated by DTX-1, 25 biological processes were found to be significantly enriched (Appendix A). The majority of these GO terms were related to transcription and biosynthesis of macromolecules, such as RNA. We also identified nine significant cellular components related to the nucleus and five significant molecular functions related to transcription. Our data suggest that genes specifically up-regulated by DTX-1 encode mainly proteins involved in the regulation of transcription. Using DAVID, the analysis outlined 13 significant terms mostly including the PI3K-Akt and MAPK signalling pathways (Appendix A). These biological pathways, through the involvement of transcriptional factors, affect the regulation of numerous processes such as cell proliferation, differentiation, inflammation, and apoptosis, in agreement with the GoMiner results. Among the major proteins involved in these signalling pathways, Ras, NFκB, MEK, and JNK are coded by genes belonging to the DTX-1 specific up-regulated cluster. The 572 genes specifically down-regulated by DTX-1 provided a list of 13 biological processes and five cellular components, without any enriched molecular function (Appendix A). Both the biological processes and the cellular components listed were mainly related to nucleosome/chromatin/chromosome assembly and to DNA replication. The additional analysis using DAVID revealed three significant terms: “mismatch repair”, “nucleotide excision repair”, and “DNA replication” (Appendix A). All included four genes coding for proteins involved in DNA replication. Our results suggest that genes specifically down-regulated by DTX-1 encode proteins related to DNA repair and replication.

For the genes specifically up-regulated by DTX-2, there were seven biological processes significantly enriched, but no cellular components and no molecular functions were depicted (Appendix A). However, no functional interaction between these GO terms was found. Using DAVID, the analysis concluded on three significant terms including the PI3K-Akt and the MAPK signalling pathways (Appendix A). Among the proteins found in these three terms, only two (p27 and transforming protein p21) are coded by genes belonging to the DTX-2 up-regulated cluster. On analysis of the 149 genes specifically down-regulated by DTX-2, only four cellular components and one molecular function were found to be significantly enriched, but no functional interaction between these GO terms was found (Appendix A). The additional analysis using DAVID revealed no significant term for the genes down-regulated by DTX-2.

### 2.3. OA-, DTX-1- and DTX-2-Induced Gene Expression

As the main modulations of gene expression were specifically regulated by DTX-1 and concerned DNA repair and MAPK signalling pathways, the expression levels of 12 key up- and down-regulated genes involved in these processes were further evaluated. Caco-2 cells were exposed to OA, DTX-1, and DTX-2 concentrations ranging from low toxicity to nearly 25% toxic (13.8 to 55.3 nM for OA, 3.5 to 13.8 nM for DTX-1, and 13.9 to 55.6 nM for DTX-2). For all genes (except *mapk8*), a concentration-dependent response of gene expression was obtained for at least one toxin (Table 1). Concerning DTX-1, among the 12 selected genes, only three genes (*rfc1*, *rfc4*, and *rpa1*) encoding proteins involved in DNA repair and replication showed significant modification of expression. For these three genes, a significant decrease of expression (29% up to 44%) compared to the control was observed at the highest DTX-1 concentration. Surprisingly, a significant decrease of expression was also observed for these three genes after exposure to DTX-2 and OA. However, even though the level of decrease was the same for DTX-1 and DTX-2, it was more pronounced with OA (between 58% and 68% compared to the control at the highest concentration). In contrast, for *rpa3*, another down-regulated gene, its expression was diminished by half compared to the control, but only after OA exposure. Concerning the seven up-regulated genes, the expression level of four of them (*ccnd1*, *rela*, *nras*, and *hras*) increased from 63% to 162% compared to the control, but only with OA. Similarly, a significant increase of expression was noted for *map2k1* (75% higher than that of the control) only after DTX-2 exposure, and for *cdkn1b* after OA exposure (63% higher than the control) and DTX-2 exposure (78% higher than the control). Lastly, the expression level of *ccnd2* was decreased, but only with OA (68% lower than that of the control).

### 2.4. Cell Cycle Arrest and MAP Kinase Pathway Quantified by High Content Analysis

The total cell number for the vehicle control was on average 1810 ± 696 cells/10 fields. After 24 h treatment, a concentration-dependent decrease in cell count was observed with the three toxins (Table 2). Consistently with the neutral red uptake assay, cells-count curves were similar for OA and DTX-2. In contrast, toxicity occurred with lower DTX-1 concentrations. A decrease in G0/G1 cells, concomitant with an increase in S and G2/M cells, was detected for the three toxins, with a maximum effect from 110.6 nM for OA, 27.7 nM for DTX-1, and 55.6 nM for DTX-2 (Figure 3). The number of phospho-H3-positive cells and phospho-MEK2-positive cells did not change significantly with moderate toxic concentrations. However, these levels increased when a strong toxic response was induced (Table 2). Finally, a significant increase in 5-ethynyl-2′-deoxyuridine (EdU)-positive cells (+139% on average) was detected only with non-toxic concentrations of OA, but without any increase in S cell numbers. We also observed a lower but non-significant increase in EdU-positive cells with DTX-1 and DTX-2.

## 3. Discussion

To investigate the toxicity of lipophilic phycotoxins on the intestine, most of the in vitro studies have used human intestinal epithelial Caco-2 cells as a model. The sub-toxic concentration tested in the transcriptomic study for the three phycotoxins was established by a cytotoxicity assay. For each toxin, a concentration-dependent decrease of viability of Caco-2 cells was observed after 24 h treatment. Other studies on undifferentiated Caco-2 cells have reported an absence of toxicity at 50 nM OA [22,24] or an IC_50_ of 50 nM [15,21], consistent with our results (IC_50_ = 75 nM). Similarly, we observed higher cytotoxicity with DTX-1 than with OA and DTX-2, as already reported on Caco-2 cells [15,24,25], but also on other cell types [14,16,26,27,28]. A sub-toxic concentration of OA (14 nM), DTX-1 (7 nM), and DTX-2 (28 nM) was chosen to investigate the specific mechanisms involved without induction of cytotoxicity.

As our goal was to identify the specific mechanisms involved for each phycotoxin, identical pathways were discarded. Therefore, our study differed from Bodero et al. [24]. Indeed, these authors analysed the transcriptomic profiles of Caco-2 cells following exposure to OA or DTX-1, but they did not discarded genes identically regulated (which represent the majority), and therefore they could not be able to distinct common regulated genes from specifically regulated genes, and so they did not reveal any difference between OA and DTX-1. Our results revealed a low number of genes up- and down-regulated only by OA, leading to few enriched biological pathways without a functional relation. The “Jak-STAT signalling pathway” was the only one identified from the genes up-regulated by OA. The inhibition of PPs by OA has been shown to enhance this pathway [29,30], whose deregulation can be associated with disorders such as cancers, immune conditions, and cardiovascular diseases [31,32]. Our results concluded that the Jak-STAT pathway was also enriched after exposure to DTX-1. In fact, this biological process is probably common to phycotoxins belonging to the OA group. Although a higher number of genes were specifically modulated by DTX-2, there were still only a few significant biological pathways identified and without a functional relation. Therefore, even though we observed a similar cytotoxic dose-response pattern for these two phycotoxins, we were not able to identify biological mechanisms that were specifically modulated by OA or DTX-2. Similarly, the same potency of OA and DTX-2 has been shown for genotoxicity in HepaRG cells [28], and for integrity of the Caco-2 cell monolayer [23]. In contrast, these two studies revealed higher toxicity of DTX-1. While a common molecular initiating event (PP1 and 2A inhibition) is certainly involved in the response of the three phycotoxins, the role of other mechanisms of action cannot be ruled out to explain the higher toxicity of DTX-1, as already suggested [3,20]. Since the inhibition potency of DTX-1 for PP5 is the highest [13], and as this PP is involved in DNA damage check point activation [33], but also inactivates Raf-1 leading to downstream signalling to MEK [34], this could contribute to the higher toxicity of DTX-1.

The specific transcriptomic profile of Caco-2 cells exposed to DTX-1 showed the greatest changes. Among the large number of genes specifically up-regulated by DTX-1, many code for proteins involved in the regulation of transcription. Most of them are related to biological pathways including the MAPK signalling pathway, which is involved in many cellular processes such as cell proliferation, differentiation, inflammation and apoptosis [35,36]. Considering the specific response of Caco-2 cells to DTX-1 treatment, we evaluated the expression of certain gene markers of mitotic block [37], and the phosphorylation of histone H3 and MEK2 protein kinase induced by MAPK pathway activation. However, we did not confirm the transcriptomic results. None of the selected genes involved in cell proliferation or in the identified signalling pathways had a fold-change above two in Caco-2 cells treated with up to 14 nM DTX-1. The response on phosphorylation of histone H3 and MEK2 protein kinase did not show any difference between DTX-1 and the other two phycotoxins. Nevertheless, irrespective of the phycotoxin, cell cycle progression was disturbed, with an increase of cells blocked in the S phase and G2/M transition phase. Similar effects were observed on Caco-2 cells and hepatic rat Clone 9 cells exposed to OA and DTX-2 [15,38,39]. However, our results showed that the effects on the cell cycle were induced with lower concentrations of DTX-1. In contrast, equipotent effects on cell cycle progression were observed with OA and DTX-2, as previously described [39].

In addition to this first analysis, we also pointed out that the genes specifically down-regulated by DTX-1 mainly code for proteins related to DNA repair and replication. Using reverse transcription quantitative polymerase chain reaction (RT-qPCR), we confirmed a slight decrease of expression, with a fold-change less than two, for the *rfc1*, *rfc4*, *rpa1,* and *rpa3* genes following exposure to the highest concentration of DTX-1. As suggested for the up-regulated genes, the strongest response obtained with microarrays assays compared to RT-qPCR assays could be explained by a difference in detection sensitivity. However, the same genes were down-regulated with OA and DTX-2 at the highest concentration, although the transcriptomic profiles for OA and DTX2 did not point out these two biological processes. Nevertheless, the transcriptomic results concluded that specific down-regulation of genes involved in DNA damage and replication by DTX-1 could be explained by the higher genotoxic potency of DTX-1 compared to OA and DTX-2, as previously reported [15,28].

In conclusion, although OA, DTX-1, and DTX-2 act through the same initiating event (PPs inhibition), DTX-1 can be distinguished by the specific modulation of gene products involved in the regulation of transcription and DNA repair. The effect on these two biological processes could explain, at least partially, the higher toxicity of DTX-1. Moreover, cell cycle disturbance occurred with lower concentrations of DTX-1, while OA and DTX-2 showed fairly equivalent toxic potential. Further studies are needed to evaluate whether other molecular mechanisms may also explain the higher toxicity of DTX-1 compared to OA and DTX-2.

## 4. Materials and Methods

### 4.1. Chemicals

Cell culture products (culture medium, non-essential amino acids, penicillin, streptomycin, and foetal calf serum (FCS)) were purchased from Gibco (Invitrogen, Cergy-Pontoise, France). OA, DTX-1 and DTX-2 standards, dissolved in methanol (MeOH) with final concentration of 17.7 µM, 17.7 µM, and 8.8 µM respectively, were supplied by the National Research Council—Institute for Marine Biosciences (Halifax, NS, Canada). Neutral red powder was purchased from Sigma-Aldrich (Saint-Quentin-Fallavier, France).

### 4.2. Cell Culture and Toxin Exposure

Caco-2 cells were obtained from the American Type Culture Collection (HTB-37, LGC Standards, Molsheim, France) and used at passages 32–40. Cells were grown in a culture medium (minimum essential medium containing 5.5 mM D-glucose, Earle’s salts, and 2 mM L-alanyl-glutamine (MEM GlutaMAX)), supplemented with 1% non-essential amino acids, 50 IU/mL penicillin, 50 μg/mL streptomycin, and 20% FCS, at 37 °C in an atmosphere containing 5% CO_2_. Caco-2 cells were seeded at 5 × 10^4^ cells/cm^2^ in 96-well plates for cytotoxicity and HCA, and in 6-well plates for microarrays and RT-qPCR assays. The day after, cells were exposed to toxins in FCS-free medium for 24 h. A vehicle control containing MeOH (5% for cytotoxicity and HCA, 0.62% for RT-qPCR, and 0.31% for microarrays) was included for each experiment. Four independent experiments were performed for cytotoxicity, HCA and RT-qPCR, and six independent experiments for microarrays were performed.

### 4.3. Cytotoxicity Evaluation by Neutral Red Uptake Assay

After 24 h of treatment with the toxins, cells were incubated for 2 h with 0.004% neutral red solution prepared in FCS-free medium before a solubilising step (acetic acid:ethanol 50% (1:99; v:v)). Absorbance was measured at 540 nm with a microplate-reading spectrofluorometer (FLUOstar OPTIMA, BMG Labtech, Champigny-sur-Marne, France). For each independent experiment (biological replicate), the median of three technical replicates was calculated, and then expressed in percentage to that of the vehicle control. The IC_50_ was determined using GraphPad Prism software (version 5.0, GraphPad Software Inc., La Jolla, CA, USA).

### 4.4. Microarray Experiments and Data Analysis

Based on cytotoxicity evaluation, Caco-2 cells were exposed for 24 h to the highest sub-toxic concentration inducing no significant alteration of cell viability: 13.8 nM OA, 6.9 nM DTX-1, and 27.7 nM DTX-2. Thereafter, total RNA was isolated using the NucleoSpin RNA II kit according to the manufacturer’s instructions with a final elution volume of 10 µL of RNase free water (Macherey-Nagel, Hoerd, France). RNA was quantified with the BioSpec-Nano (Shimadzu, Marne la Vallée, France), and RNA integrity was assessed with the “Experion RNA StdSens analysis” kit using the Experion automated electrophoresis system (Bio-Rad, Marnes-la-Coquette, France). Only RNA with an RNA quality indicator ≥ 9 was used for further experiments (Experion software 3.0; Bio-Rad). A negative extraction control of lysis buffer RA1 was included for contamination assessment. Probe preparation and hybridisation (using 4 × 44K Whole Human Genome 70-mer oligo-chips, G4112F, Agilent Technologies, Massy, France), with a completely randomised design were performed at the ANSES transcriptomic platform. Labelling was undertaken with either cyanine-3 CTP or cyanine-5 CTP, and absorbance was measured at 532 nm (for cyanine-3-labelled cRNA samples) or 635 nm (for cyanine-5-labelled cRNA samples) [40].

Raw data extraction, quality control, and Lowess normalisation were performed as previously described [40]. A background signal was calculated from the mean of the one hundred lowest values of each sample. Following this, the normalised data were filtered on threshold intensity (3 times the background signal): for each probe, values were selected if the median value for at least one experimental condition was higher than the threshold intensity. The dataset, including the 26 960 post-filtering probes, was labelled “filtered data” and deposited in the National Center for Biotechnology Information (NCBI) Gene Expression Omnibus database through the accession number GSE159293. For each toxin, from the “filtered data”, the differentially expressed genes compared to the vehicle control were selected at *p* < 0.05 (Student *t*-test) and with a fold change (FC) greater than two (for “up-regulated genes”) or less than 0.5 (for “down-regulated genes”). Among these genes, those specifically altered by only a single toxin (and not by the other two) were distinguished, leading to six clusters: for each toxin, “specific up-regulated genes”, and “specific down-regulated genes”. An analysis of these clusters was performed using the GoMiner tool (http://www.discover.nci.nih.gov/gominer/index.jsp). GO terms with a false discovery rate (FDR) score below 0.05 and an enrichment score above 1 were considered significant. An additional analysis for the same clusters was performed with the DAVID bioinformatics resource (http://david.abcc.ncifcrf.gov). Using the DAVID Functional Annotation Tool, we visualised genes on Kyoto Encyclopedia of Genes and Genomes (KEGG) pathway maps, selecting terms with a *p*-value below 0.05.

### 4.5. RT-qPCR

After 24 h treatment of Caco-2 cells with concentrations ranging from low toxicity to near 25% toxic (OA 13.8 to 55.3 nM; DTX-1 3.5 to 13.8 nM; DTX-2 13.9 to 55.6 nM), total RNA was isolated, quantified and assessed for its integrity, as described above. A negative extraction control was included for the contamination assessment. We applied the guidelines for qPCR assay design and reporting [41]. Reverse transcription (RT) was performed with 2 µg of total RNA using the High Capacity RNA-to-cDNA kit (Applied Biosystems, Foster City, CA, USA) according to the manufacturer’s instructions. Reaction volume was set to 20 µL and RT was performed at 37 °C for 60 min prior to a stopping step for 5 min at 95 °C. Negative controls were included [42].

For primers, their design and the in-silico analyses of their specificity were performed together, using the *Basic Local Alignment Search Tool* (BLAST) for primers (http://www.ncbi.nlm.nih.gov/BLAST/) with, for each gene, at least one primer designed to span an exon-exon junction. All primers were purchased from Sigma-Aldrich (Lyon, France), and additional information regarding the target genes and oligonucleotide primers is listed (Appendix A). Using NormFinder software (version 0.953; Aarhus, Denmark), the *gapdh* gene was chosen as the reference gene since it did not exhibit any significant variation in expression among all the samples.

Quantitative PCR was performed on a Chromo4Real-Time Detector in low-profile 8-white tubes strips (Bio-Rad). SYBR Green chemistry was used. Reactions were performed on three technical replicates in a total volume of 10 µL containing 1X Power SYBR GREEN PCR Master Mix (Applied Biosystems, Foster City, CA, USA), 300 nM each primer, and 0.2 or 0.4 ng cDNA for moderate or poorly expressed genes respectively. Negative quantitative PCR controls of RNase-free water were included in each run for contamination assessment. The thermal cycling conditions were 94 °C for 10 min, followed by 40 cycles of denaturation at 94 °C for 15 s, annealing at the determined temperature for 15 s, and polymerisation at 72 °C for 30 s. Opticon Monitor software (version 3.0; Bio-Rad) was used for the quantitative analysis, and melting curve analysis was used to check the specificity of each amplicon. Threshold Cqs were calculated from a baseline subtracted curve fit. For contamination assessment, the results revealed that the ΔCqs of the samples were at least 5 compared to the various controls (extraction, RT, no-reverse transcriptase, and qPCR). Calibration curves were established for each gene from a serial two-fold dilution of a reference sample (pool of cDNA samples). Using these calibration curves, for each sample, median relative amounts of mRNA of the target genes were calculated and then normalised to that of the reference gene, *gapdh*. These normalised medians were used for statistical analyses and values are presented as arbitrary units.

### 4.6. HCA of Multiparametric Toxicity Endpoints

Following 24 h treatment, cells were fixed for 10 min with 4% paraformaldehyde in phosphate buffered saline (PBS), and permeabilised for 10 min with 0.2% Triton X-100. Cells were then incubated for 30 min in blocking solution (PBS with 1% BSA and 0.05% Tween-20) and successively incubated at room temperature for 2 h and 1 h with, respectively, primary and secondary antibodies prepared in blocking solution. Antibodies were purchased from Abcam (Cambridge, UK) and used as follows: 0.5 µg/mL Rabbit Anti-Histone H3 (phospho S10) antibody (ab5176), 2 µg/mL Rabbit Anti-MEK2 (phospho T394) antibody (ab30622), 1 µg/mL Goat Anti-Rabbit IgG (H&L) Alexa Fluor 647 antibody (ab150079), and 1 µg/mL Goat Anti-Rabbit IgG (H&L) Alexa Fluor 488 antibody (ab150077). For nuclear identification, cells were incubated for 5 min with DAPI 1 µg/mL/0.05% Tween in PBS Immunostaining with EdU was performed 30 min before the end of phycotoxin exposure using a Click-iT^TM^ Plus EdU Cell Proliferation kit, according to the manufacturer’s instructions (Thermo Fisher Scientific, Courtaboeuf, France, C10637). Plates were scanned with a Thermo Scientific Arrayscan VTI HCS Reader (Thermo Fisher Scientific), and 10 fields (size of each field: 660 × 660 µm) were analysed per well. Using nuclear DAPI staining and the Cell Cycle Analysis module of BioApplication software, cells were classified in the different cell cycle phases and expressed in relation to the total cell number. Phospho-histone H3 and EdU (for cell cycle progression), and phospho-MEK2 (for the MAP kinases pathway) were quantified in the nuclei using the Target Activation module. Cells were defined as positive when their average intensity exceeded a threshold of two standard deviations compared to the average intensity of the vehicle control. The percentages of positive cells were expressed in relation to the total cell numbers. For each independent experiment, the mean of two technical replicates (wells) was calculated and used for statistical analyses.

### 4.7. Statistical Analysis

Statistical analyses were performed using GraphPad Prism software (version 5.0). For cell counts, data were analysed using the one-sample *t*-test, with “100” as the theoretical mean. Means were considered significantly different from 100 at *p* < 0.05. For IC_50_ values, RT-qPCR results, and HCA data, the homogeneity of the variances was verified. Thereafter, an analysis of variance was performed. When the concentration effect was significant (*p <* 0.05), the values were compared. For IC_50_, values were compared with each other using Bonferroni’s test. For RT-qPCR and HCA, values were compared to the vehicle control using Dunnett’s test. Differences were considered significant at *p* < 0.05. The values presented are means ± SEM.

## Figures and Tables

**Figure 1 toxins-12-00783-f001:**
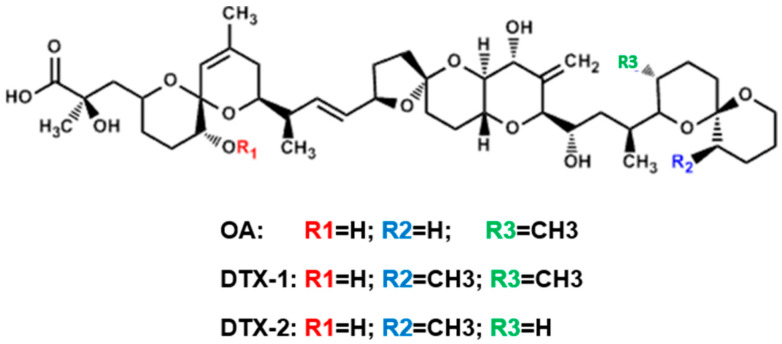
Structure of okadaic acid (OA), dinophysistoxin-1 (DTX-1), and dinophysistoxin-2 (DTX-2) [15].

**Figure 2 toxins-12-00783-f002:**
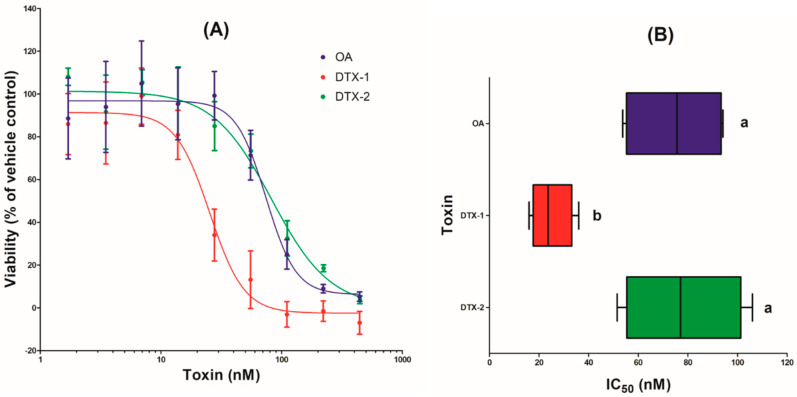
Cytotoxicity in Caco-2 cells after 24 h exposure to okadaic acid (OA), dinophysistoxin-1 (DTX-1), and dinophysistoxin-2 (DTX-2). Cytotoxicity was measured by the neutral red uptake assay. Four independent experiments were performed. (**A**) Viability values are presented as means ± SEM and expressed as percentages of the vehicle control. (**B**) IC_50_ values are presented as box plot. ^a, b^: box plot without a common letter differ (*p* < 0.05).

**Figure 3 toxins-12-00783-f003:**

Cell cycle progression in Caco-2 cells after 24 h exposure to phycotoxins. (**A**) okadaic acid (OA); (**B**) dinophysistoxin-1 (DTX-1); (**C**) dinophysistoxin-2 (DTX-2). The cells were classified in the different cell cycle phases using nuclear DAPI staining (>G2/M corresponds to polyploid cells). Values are presented as means ± SEM and expressed in relation to the total cell number. Four independent experiments were performed. *: significantly different from the vehicle control (*p* < 0.05).

**Table 1 toxins-12-00783-t001:** Relative gene expression in Caco-2 cells after 24 h exposure to okadaic acid (OA), dinophysistoxin-1 (DTX-1), and dinophysistoxin-2 (DTX-2). The genes were chosen from the transcriptomic analysis. Values are presented as means ± SEM and were normalised to the *gapdh* reference gene. Four independent experiments were performed. ^a^
*p* < 0.05, ^b^
*p* < 0.01 and ^c^
*p* < 0.005: significantly different from the vehicle control.

Gene	Control	OA1 3.8 nM	OA 27.7 nM	OA 55.3 nM	DTX-1 3.5 nM	DTX-1 6.9 nM	DTX-1 13.8 nM	DTX-2 13.9 nM	DTX-2 27.8 nM	DTX-2 55.6 nM
*ccnd1*	0.18 ± 0.01	0.25 ± 0.03	0.35 ± 0.01 ^b^	0.31 ± 0.01	0.19 ± 0.01	0.24 ± 0.01	0.27 ± 0.01	0.20 ± 0.01	0.26 ± 0.01	0.26 ± 0.01
*ccnd2*	0.28 ± 0.02	0.32 ± 0.01	0.38 ± 0.08	0.09 ± 0.01 ^b^	0.26 ± 0.04	0.29 ± 0.03	0.26 ± 0.02	0.30 ± 0.02	0.27 ± 0.02	0.15 ± 0.05
*rela*	0.99 ± 0.17	0.96 ± 0.05	0.96 ± 0.16	2.60 ± 0.26 ^c^	0.83 ± 0.13	0.86 ± 0.09	1.28 ± 0.17	0.86 ± 0.11	1.02 ± 0.13	1.44 ± 0.25
*nras*	0.48 ± 0.05	0.48 ± 0.02	0.54 ± 0.07	0.86 ± 0.05 ^c^	0.39 ± 0.04	0.45 ± 0.03	0.51 ± 0.05	0.45 ± 0.04	0.44 ± 0.04	0.60 ± 0.08
*map2k1*	0.64 ± 0.12	0.46 ± 0.03	0.54 ± 0.12	0.93 ± 0.20	0.69 ± 0.07	0.59 ± 0.04	0.54 ± 0.02	0.48 ± 0.06	0.63 ± 0.09	1.13 ± 0.23 ^a^
*mapk8*	0.39 ± 0.02	0.34 ± 0.04	0.35 ± 0.04	0.52 ± 0.03	0.39 ± 0.03	0.41 ± 0.03	0.38 ± 0.05	0.33 ± 0.03	0.40 ± 0.05	0.48 ± 0.01
*cdkn1b*	0.34 ± 0.04	0.30 ± 0.03	0.32 ± 0.05	0.55 ± 0.07 ^a^	0.38 ± 0.02	0.38 ± 0.04	0.44 ± 0.01	0.31 ± 0.03	0.42 ± 0.08	0.60 ± 0.06 ^b^
*hras*	0.38 ± 0.03	0.47 ± 0.04	0.53 ± 0.04	0.62 ± 0.03 ^b^	0.39 ± 0.06	0.41 ± 0.05	0.52 ± 0.03	0.37 ± 0.02	0.46 ± 0.05	0.52 ± 0.05
*rfc1*	0.79 ± 0.16	0.66 ± 0.03	0.48 ± 0.06 ^a^	0.33 ± 0.01 ^c^	0.68 ± 0.05	0.58 ± 0.05	0.45 ± 0.03 ^b^	0.55 ± 0.06	0.45 ± 0.02 ^b^	0.45 ± 0.04 ^b^
*rfc4*	0.63 ± 0.05	0.53 ± 0.04	0.46 ± 0.05	0.20 ± 0.02 ^c^	0.56 ± 0.05	0.59 ± 0.07	0.35 ± 0.03 ^b^	0.51 ± 0.04	0.44 ± 0.05	0.37 ± 0.07 ^b^
*rpa1*	0.58 ± 0.05	0.53 ± 0.07	0.50 ± 0.02	0.23 ± 0.01 ^c^	0.49 ± 0.04	0.54 ± 0.02	0.41 ± 0.01 ^a^	0.52 ± 0.06	0.43 ± 0.04	0.38 ± 0.05 ^a^
*rpa3*	0.50 ± 0.03	0.49 ± 0.03	0.46 ± 0.05	0.25 ± 0.03 ^b^	0.55 ± 0.04	0.51 ± 0.02	0.41 ± 0.01	0.62 ± 0.05	0.53 ± 0.09	0.38 ± 0.06

**Table 2 toxins-12-00783-t002:** High content analysis of toxicity endpoints in Caco-2 cells after 24 h exposure to okadaic acid (OA), dinophysistoxin-1 (DTX-1), and dinophysistoxin-2 (DTX-2). Cell count values are expressed as percentages of the vehicle control. Phospho-H3-, EdU- and MEK2-positive cells were expressed as percentages of the total cell numbers (see Materials and Methods). Values are presented as means ± SEM. Four independent experiments were performed. ^a^
*p* < 0.05, ^b^
*p* < 0.01 and ^c^
*p* < 0.005: significantly different from the vehicle control.

**OA**	**Control**	**1.7 nM**	**3.5 nM**	**6.9 nM**	**13.8 nM**	**27.7 nM**	**55.3 nM**	**110.6 nM**	**221.3 nM**	**442.5 nM**
Cell count	100 ± 0.0	94.2 ± 5.1	97.8 ± 6.0	86.6 ± 4.2 ^a^	73.2 ± 8.4 ^a^	48.8 ± 11.2 ^c^	22.8 ± 6.9 ^c^	16.3 ± 3.5 ^c^	11.2 ± 4.0 ^c^	6.8 ± 1.6 ^c^
Phospho-H3	2.0 ± 0.0	2.2 ± 0.2	2.2 ± 0.5	2.2 ± 0.5	3.2 ± 0.9	8.5 ± 1.8	13.2 ± 3.1	12.0 ± 3.6	14.5 ± 5.5	19.0 ± 6.3 ^b^
EdU	5.7 ± 0.2	14.0 ± 1.2 ^b^	13.2 ± 2.7 ^a^	13.7 ± 2.2 ^b^	9.7 ± 2.4	7.2 ± 1.8	0.2 ± 0.2	0.5 ± 0.3	0.5 ± 0.5	0.0 ± 0.0
Phospho-MEK2	5.0 ± 0.0	5.2 ± 1.1	5.5 ± 1.6	5.0 ± 1.4	8.0 ± 2.7	6.0 ± 2.0	12.5 ± 3.3	20.0 ± 4.6	29.0 ± 8.3 ^b^	31.2 ± 9.2 ^b^
**DTX1**	**Control**	**1.7 nM**	**3.5 nM**	**6.9 nM**	**13.8 nM**	**27.7 nM**	**55.3 nM**	**110.6 nM**	**221.3 nM**	**442.5 nM**
Cell count	100 ± 0.0	106.2 ± 3.2	90.9 ± 5.9	75.5 ± 7.5 ^a^	46.5 ± 9.9 ^c^	23.0 ± 6.6 ^c^	19.2 ± 5.8 ^c^	21.1 ± 5.2 ^c^	15.4 ± 5.4 ^c^	7.8 ± 2.5 ^c^
Phospho-H3	2.0 ± 0.0	2.7 ± 0.3	3.8 ± 0.9	4.8 ± 1.6	8.3 ± 2.4	9.4 ± 4.2	8.6 ± 2.7	13.1 ± 6.5	21.5 ± 13.5	16.3 ± 9.2
EdU	5.7 ± 0.2	9.2 ± 0.7	5.0 ± 2.7	9.4 ± 6.2	2.0 ± 1.4	0.2 ± 0.1	0.1 ± 0.1	0.2 ± 0.1	0.2 ± 0.2	0.0 ± 0.0
Phospho-MEK2	5.0 ± 0.0	9.8 ± 2.4	5.7 ± 1.2	9.0 ± 2.9	13.9 ± 4.6	16.8 ± 2.7	13.2 ± 4.1	18.3 ± 3.9	22.1 ± 10.7	42.4 ± 14.0 ^b^
**DTX2**	**Control**	**1.7 nM**	**3.5 nM**	**7 nM**	**13.9 nM**	**27.8 nM**	**55.6 nM**	**111.3 nM**	**222.5 nM**	**445 nM**
Cell count	100 ± 0.0	88.9 ± 7.1	84.3 ± 6.8	81.4 ± 5.0 ^b^	66.3 ± 9.1 ^b^	42.3 ± 8.9 ^c^	24.2 ± 6.9 ^c^	18.2 ± 4.7 ^c^	22.7 ± 6.4 ^c^	16.5 ± 4.2 ^c^
Phospho-H3	2.0 ± 0.0	3.0 ± 0.4	2.5 ± 0.1	2.5 ± 0.5	6.1 ± 3.1	7.8 ± 3.1	10.0 ± 3.3	13.0 ± 2.9	22.1 ± 10.4 ^a^	15.3 ± 7.2
EdU	5.7 ± 0.2	4.1 ± 2.0	2.7 ± 1.5	9.1 ± 5.1	7.7 ± 5.9	0.7 ± 0.6	4.5 ± 4.1	3.5 ± 3.1	0.1 ± 0.1	0.1 ± 0.1
Phospho-MEK2	5.0 ± 0.0	4.3 ± 0.8	4.6 ± 1.3	5.2 ± 0.4	5.8 ± 2.0	7.7 ± 2.1	12.3 ± 3.9	14.1 ± 3.1	13.8 ± 4.3	19.2 ± 5.3 ^a^

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
