# Peer review of "Differences in Toxic Response Induced by Three Variants of the Diarrheic Shellfish Poisoning Phycotoxins in Human Intestinal Epithelial Caco-2 Cells"

_toxins, 2020, doi:10.3390/toxins12120783_

Round 1

Reviewer 1 Report

Manuscript ID: toxins-1010864 Differences in Toxic Response induced by three Variants of the Diarrheic Shellfish Poisoning Phycotoxins in Human Intestinal Caco-2 Cells

Abstract

Line 13 “human intestinal epithelial Caco-2 cells…”

Line 31 “… are produced by certain species belonging to the genera Prorocentrum and Dinophysis”.

Results

Lines 78-81 belong in the methods.

Line 124. Delete sentence “The results are summarised in Table 1. Any reference to a table which contains results always goes at the end of the results sentence in brackets (Table 3). Please change throughout manuscript.

Table 1 Could significant result be bolded/highlighted rather than superscripts which are hard to read?

Table 1 & Table 2 Where 4 independent experiments performed for each toxin concentration? Be specific here and in the methods.

Figure 2 Cannot read the legend at the bottom of each graph.

Line 170 total cell number (=?)

Discussion

Lines 187-188 belong in methods

I am wondering if there should be some discussion on the implication of this work in a broader sense – regulatory limits for these analogues, suitability and sensitivity of detection limits around these analogues in relation to the findings of this work.

Methods

Section 4.1 Please provide more details on cell culture products and certified reference material eg lot/batch no.s; concentrations etc.

Line 250 Earle’s salts – provide supplier/producer

Line 252 How was the atmosphere of 5% CO2 maintained?

Line 256 Please clarify “At least four independent experiments for cytotoxicity, HCA and TR-qPCR, and up to six independent experiments for microarrays were performed”. How many exactly were performed for each?

Line 259 Relabel heading to include ‘Cytotoxicity’

Line 264 What is meant by “expressed in relation to that of the vehicle control”? Was the vehicle control subtracted from each replicate, then the median determined? Why was the median used and not the mean of the replicates? Be specific.

Line 269 How was “no significant alteration of cell viability” determined? Please include detail.

Line 270 and 275 It is not enough to say “as previously described [40]”. Please provide some methodology, at least, in brief

Line 271 “assessed for its integrity” is vague. Please be specific on how this was done – what criteria/test/equipment etc.

line 290-291 how were these toxin concentrations derived/selected?

Line 291 “isolated, quantified and assessed for its integrity as described above” but you have not provided this information above…

Line 294 Again, methods need to be included here, even in brief.

Line 302 0.2 or 0.4 depending on the target gene – how was this determined?

Line 309 “as previously described”, again more detail is necessary.

Line 330 Did the data meet the assumptions of an ANOVA? Please confirm that data was normally distributed, distributions have the same variance and the data are independent.

Line 331 When was a Bonferroni test used and when was a Dunnett’s test used? Why were different tests used? Please be specific and ensure all statistical results identify which test was used for each.

Reviewer 2 Report

Marine toxins have a multitude of effects, likely due to different MOAs.  However, only the "typical" MOAs are usually studied.  This paper seeks to discover additional factors that contribute to toxicity, especially by alternative means, which is greatly needed to understand the full ramifications of marine toxins in human health.  Overall, it is well written and executed, and it is an important contribution to the understanding of OA/DTX toxicity.  I have minor comments.

Intro:

 - Line 41 is a little confusing in light of the results.  DTX-2 is the "highest" (IC50) for PPs.  Please clarify if this means binding, activity, or cytotoxicity (I think you mean activity/inhibition, but it could mean binding, which doesn't always translate directly to inhibition).  That will become important once the results are explained because DTX-1 is higher in cytotxicity and has a higher IC50. 

 - Also, please provide info on general cytotoxicty that's found in the literature for these three toxins. It is in the discussion, but it would help clarify the point above to also include it in the intro with a bit more explanation.

 - Since OA and DTXs are structural analogs with slight differences that contribute to differences in toxicity, please provide a structure figure to illustrate those differences for the reader.

Results:

 - Please explain "filtered data" in line 75.

 - Table 1 is a little difficult to follow.  Is there a better way to represent this data (dot/line plot or bar chart)?  Also, this data compares to VC, but it seems as if it should compare within the toxins to find differences in MOAs among them.

 - The discussion explains data differences among toxins (e.g. Jak/Stat) that are not necessarily demonstrated in the data tables/graphs of the manuscript.  These important differences should be illustrated as primary figures/tables, not as supplementary data.  Supplementary data is fine for the massive amount of data generated by a study like this, but key data should be presented in the results section.  Or, if it is implied that the genes illustrated in table 1 are that, then that link needs to be better explained.  Right now, it appears table 1 indicates DNA repair and MAPK pathways.

Discussion

 - Please explain the importance of pre-filtering step (line 185) to illustrate the difference from this study and Bordero.  What is lost with pre-filtering?  Also, this difference is unclear considering the results and methods say the data was "filtered data".

 - The intro (previous literature) establishes an order of toxicity/activity that DTX-2 is greater than OA.  The difference with DTX-1 and OA/DTX-2 can certainly be explained by the transcriptomic profiles from the results, but the authors found that toxicity for DTX-1 = OA, and there was nothing enlightening about the transcriptomic profiles between DTX-1 and OA.  So, how do the authors explain the roughly equivalent toxicity and IC50 for DTX-2 and OA found in this study?

Methods

- please specifically define "up to" 4 or 6 independent experiments, as that could mean as little as 1 (which would not be enough).  
